# Model-Based Analysis and Improvement of Vehicle Radiation Emissions at Low Frequency

**Feng Gao** [1,2,*] **, Qing Wang** [2] **and Yu Xiong** [3]

1. Chongqing EMC Engineering Technology Research Center, Chongqing 400044, China
2. College of Mechanical and Vehicle Engineering, Chongqing University, Chongqing 400044, China; 201934131013@cqu.edu.cn
3. Dongfong Liuzhou Motor Co., Ltd., Liuzhou 545005, China; xiongy@DFLZM.COM
* Correspondence: gaofeng1@cqu.edu.cn; Tel.: +86-189-9618-8196

**Abstract:** With the development of electrification and intelligence, the electromagnetic environment of intelligent and electric vehicles becomes complicated and critical because of the high voltage/current of power components, the computation units with high frequency and the dense radio systems. These pose great challenges for the design of vehicle radiation emissions. To improve the development efficiency, a model-based analysis and improvement strategy is proposed. Firstly, a topological approach is presented to decouple and model the vehicle-level radiation problem. By this topological model, each technical factor is analyzed from both of its contribution and sensitivity to the radiation emission, which are further integrated together using the entropy weight method to generate the technical evaluation score. Then, other untechnical factors, i.e., the cost and application difficulty, are further combined with the technical evaluation results by the analytic hierarchy process to determine the final solution. This strategy has been applied to solve a radiation problem of an electric vehicle at low frequency to validate its effectiveness and show some application details.

**Keywords:** electromagnetic compatibility (EMC); electric vehicle; model based analysis; radiation emission

## 1. Introduction

Electrification and intelligence are the main technical trends of vehicle industry because of their contributions on traffic safety, energy conservation and emission reduction of air pollutant [1,2]. From the viewpoint of electromagnetic compatibility (EMC), the electromagnetic environment inside vehicles deteriorates substantially due to the high voltage/current of electrical drivetrains, dense radio systems and onboard electronics with high frequency/speed [3]. Meanwhile the increase of potential interference sources, sensitive equipment and coupling paths poses great challenges on the design efficiency of vehicle-level EMC [4]. Simulation is an effective way to analyze EMC, which can be adopted throughout the whole vehicle product development process. However, when using the simulation method, its effectiveness depends heavily on the model accuracy. The key of the EMC modelling is to characterize the equivalent interference sources and sensitive equipment [5], and map the propagation paths of interferences among them [6].

The modeling method normally is selected according to the electrical size. Electrically small parts, whose structure is much less than the wavelength of the considered signals, can be regarded as a circuit and the propagation of electromagnetic waves in its physical-space can be ignored. For example, Jeschke et al. extracted the electrical characteristics of a drive inverter to create an equivalent lumped circuit and a good simulation accuracy was achieved [7].

Otherwise, the propagation of electromagnetic wave in electrical large parts, such as cables, should be considered in detail. Some approximation methods have been proposed for such structures, which are integrated in the numerical solving software for 3D

electromagnetic fields. For instance, to acquire a reliable model of large electrical cables for the prediction of the radiated emission at high frequency, the distributed parameters of automotive cables were converted into the lump parameters by the partial element equivalent circuit method [8]. Moreover, the transfer function was also used to describe the propagation characteristic of the radiated coupling path [9]. This approach is concise and explicit, but the transfer function varies with its port impedance. To overcome this limitation, Gao et al. adopted the multi-port network with Z-parameter to describe the electromagnetic propagation path of the high voltage components in an electric vehicle (EV) [10]. The coupling of the high and low power systems in the power distribution unit (PDU) is further considered in [11]. To reduce the scale of the model, electromagnetic topology is an effective way, which decomposes a large and complex system into multiple subsystems according to the electromagnetic shielding level [12,13]. This method has been successfully applied to evaluate the EMC of Boeing 707 airplanes [14].

With the help of simulation models, the comparative analysis of the improvement of EMC between different methods becomes feasible. For example, Kamruzzaman et al. compared by simulation the reduction effect of interferences when using different MOSFET and IGBT switches [15]. Unfortunately, we require the candidate solutions beforehand when using this approach. This depends greatly on the engineer's experience. For systems as complex as the vehicle-level radiation emissions, the key coupling paths, interference sources and sensitive equipment are hard to determine directly, let alone the possible solutions. Moreover, the vehicle radiation emission is a systematic problem. Reducing the source interference and weakening the transfer path both are beneficial to the vehicle radiation emission. Furthermore, how to make a balance between these technical factors, the application difficulty and costs systematically is still a problem too [16].

To overcome the design challenge of vehicle-level radiation emissions and improve the development efficiency, a model based analysis and improvement strategy is proposed. Firstly, a topological modelling process is suggested to decouple the concerned problem into multiple networks to reduce the modelling difficulty and consumption of computation resources. Then based on the topological model, an evaluation score is obtained for each technical factor by the entropy weight method (EWM) according to the contribution and sensitivity analysis results. Finally, other non-technical factors, i.e., cost and application difficulty, are further combined with the technical ones by using the analytic hierarchy process (AHP) to determine the final solution. To validate the effectiveness and show some application details of this new strategy, it has been applied to the analysis and improvement of the radiation problem of an EV at low frequency.

The following paper is organized as follows: The model based diagnosis and improvement strategy is introduced in Section 2 based on the problem analysis of the development for vehicle level radiated emissions; Sections 3 and 4 expound the topological modelling method, and the model based diagnosis and improvement approach, respectively; The proposed strategy is applied to solve a radiation problem of EV at low frequency in Sections 5 and 6 concludes the paper.

## 2. Problem Description

An electric and intelligent vehicle is composed of various types of electrical/electronic equipment connected by complicated cables in a compact space, and faced with uncontrollable electromagnetic environments because of its mobility. These pose great challenges on the modeling, analysis and improvement of vehicle-level radiation emissions because:

### 2.1. Simultaneous Existence of Small and Large Electrical Parts

At the vehicle level, the size of such parts as vehicle body and cable is normally much greater than the minimum wavelength of the interferences of concern. The 3D numerical method is required to investigate their influences on the electromagnetic field. On the contrary, if the module size is much smaller, it can be considered as a lumped system. However, current commercial software for solving 3D electromagnetic field only

can integrate simpler circuits, let alone the physical process of other fields, such as logic, mechanical and etc. Moreover, enormous mesh grids are required for the physical model of a detailed modular integrated with a vehicle.

### 2.2. Complexity of Potential Factors, Coupling Paths and Failure Modes

In an EV, various electrical/electronic components and many metal structures are assembled in a compact space. Near field coupling is widespread and how to extract the equivalent problem to be modeled greatly depends on the engineer's experience. Otherwise, the simulation model may become too complicated to be solved. Moreover, a vehicle radiation problem may be activated by multiple factors simultaneously. Even with a precise model, it is inconvenient and time consuming to evaluate vehicle radiation emissions under different combinational conditions to find out the main sources and determine the final solution.

### 2.3. Balance between Performance, Engineering Application and Costs

Even if the main factors contributing to the vehicle radiation problem are identified technically, it is still not enough to determine the final solution in production because this also depends on their application difficulties and costs. Such untechnical factors are hard to evaluate quantitatively and the final decision is mostly made based on subjective experience. This may cause an ineffective design and increased validation costs. To deal with the aforementioned difficulties, the proposed process for model-based diagnosis and development of vehicle radiated emissions is depicted by Figure 1.

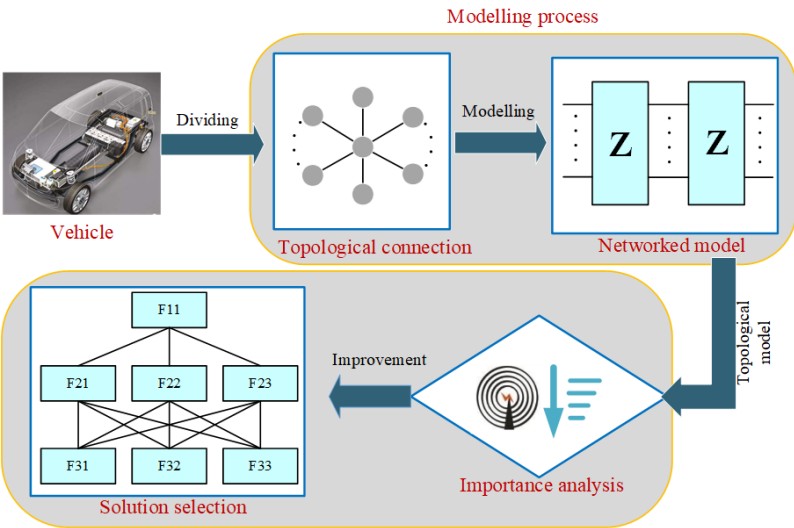

**Figure 1.** Model based analysis and improvement process.

The advantages of this process are:

(1) The complete vehicle radiation problem is divided into multiple submodels by:
  (a) The shielding effect of such metal structures as vehicle body, component shell, etc.;
  (b) The electrical size of the modeled components.
(2) The modelling difficulty of each subsystem is comparatively reduced. Moreover, different subsystems can be modelled by different methods. How to decouple the vehicle radiation problem is discussed in Section 3.1.
(3) The multi-port network characterized by Z-parameter is adopted to describe the internal couplings of subsystems caused by the near field and electromagnetic radiation. It is not necessary for the engineer to extract the equivalent vehicle radiation model exactly. The final interference is obtained by solving an algebraic equation including

the functions of multi-port networks and their connection topologies. Compared with the 3D numerical method, this reduces the requirement of computational resources greatly. The topological model for vehicle-level radiated emissions is introduced in Section 3.2.

(4)   The influences of different technical factors on radiated emissions are evaluated quantitatively and directly base on the model by considering their sensitivities and contributions. The engineer need not provide possible improvement measures in advance. The details of the evaluation process can be found in Section 4.1.

(5)   The tree structure is suggested to model all factors including both technical and untechnical ones, which influence the determination of final solutions. Only the comparative importance among different factors belonging to the same category need to be evaluated by the engineer. This is easier to implement in practice and reduces the requirement of engineer's experience. The model-based improvement process is discussed in Section 4.2.

## 3. Topological Modelling Process

Firstly, to reduce the difficulty of modelling, the decoupling principles of vehicle-level radiated emissions are discussed in Section 3.1. Then in Section 3.2 a global multi-port network model is further established by connecting all subsystems together, which are described by their impendence parameters to remove the influence of the port's characteristic.

### 3.1. Decoupling of Vehicle Radiation Problem

In essence, the objective of decoupling is to decompose the entire problem into several sub-ones. Each subsystem is comparatively easier to be modelled by the appropriate method. The following vehicle's characteristics make it suitable to use the decoupling method:

(1)   Most components are electrical small and wrapped in metal shells, which are well grounded. Such components can be considered as lumped parameter systems when studying the vehicle-level radiation emissions.

(2)   Vehicle manufactures focus on the system design, while the detailed design of components is carried out by the supplier. Considering both the demand for secrecy and the concerns of different roles, a decoupling structure is required to make each supplier responsible for their own parts.

(3)   Though there exist nonlinear materials and devices in components, their connection cables and surroundings are linear. These linear systems with their internal couplings can be described by such methods as the multi-port network, transfer function, etc.

Accordingly, the suggested decoupling process is depicted in Figure 2, where the subsystems are labeled as $V_{i,j}$.

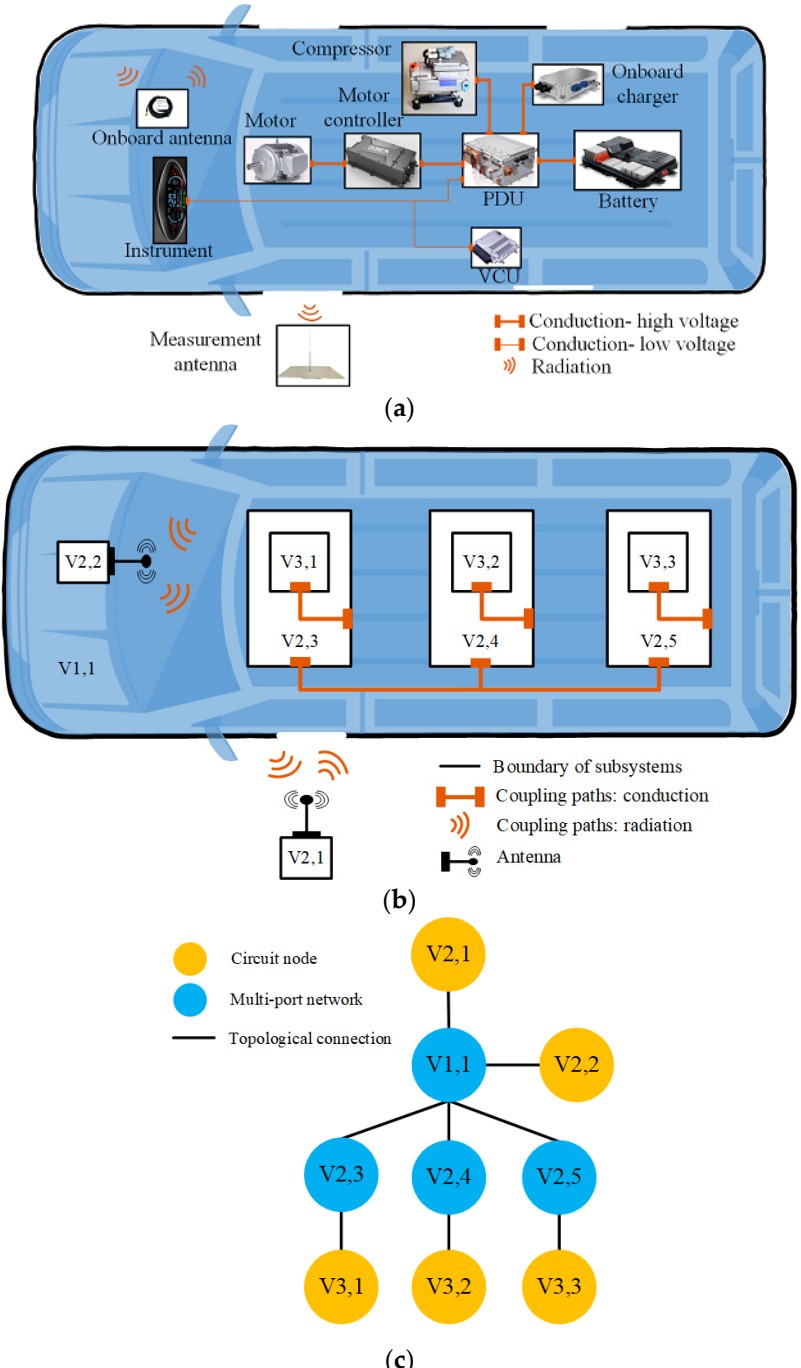

**Figure 2.** Decoupling process of vehicle radiation problem: (**a**) Original system; (**b**) Equivalent system; (**c**) Topological connection.

The principles for the conversion of the original vehicle-level radiation problem to a topological model are:

(1) The parts without direct radiation coupling with others can be considered as a separate subsystem. The direct radiation coupling means that interferences flow into and out of a part through space. At this condition, it is hard to separate the propagation of electromagnetic wave further. On the contrary, if there is no such direct radiation coupling, the part can be considered as a separate subsystem. For example, in Figure 2b, since the boundaries are metal materials which shield the radiation effectively, $V2,3 \sim V2,5$ and $V3,1 \sim V3,3$ only receive the conduction interference

and there is no direct radiation coupling among them. As a result, they are considered as separated subsystems as shown in Figure 2c.

(2) Conduction and radiation coupling paths that belong to the same volume should be integrated into one network. As shown in Figure 2b, the transfer paths between V2, 3 ∼ V2, 5 are conduction couplings in the volume V1, 1. But in the same volume, there also exist other radiation couplings. These radiation and conduction paths should be integrated into one network, i.e., V1, 1 in Figure 2c, unless it is ensured that the coupling among the radiation and conduction paths is negligible. By this way, the requirement of prior knowledge about the radiation problem is reduced. Meanwhile, each subsystem can be modeled separately and the radiated emissions can be solved by solving an algebraic equation (see Section 3.2). The consumption of calculation resources is greatly reduced.

(3) Electrical large and small subsystems are described by multi-port networks and circular nodes, respectively. The subsystems are denoted by circles in Figure 2c, and the lines represent their subordinate relationships. For an electrical large subsystem, the interference interactions among different devices are very complicated and it is difficult to find out that required to be modeled properly. To reduce the requirement of engineer's experiences and the priori knowledge of the concerned problem, the multi-port network is adopted to describe electrical large systems denoted by the blue circles in Figure 2c. It includes all possible coupling paths and is easily to be solved numerically. The electrical small subsystems are modeled by circuit nodes (Yellow circles in Figure 2c) connected to the multi-port network. Particularly, the subsystems acting as the interference sources or sensitive equipments have to be modeled as circuit nodes.

It should be noted that the determination of interference sources or sensitive equipment depends on the type of studied problems. For example, the on-board equipment serves as the interference source and the measurement antenna act as the sensitive equipment for a vehicle radiation problem. On the contrary, for a susceptibility problem, the radiation antenna acts as the interference source and the onboard units become to be the sensitive equipment.

*3.2. Global Network Model*

Based on the topological model introduced in Section 3.1, the algebraic function is established to calculate interferences induced on the sensitive equipment by combining all sub-systems into one global network as shown in Figure 3.

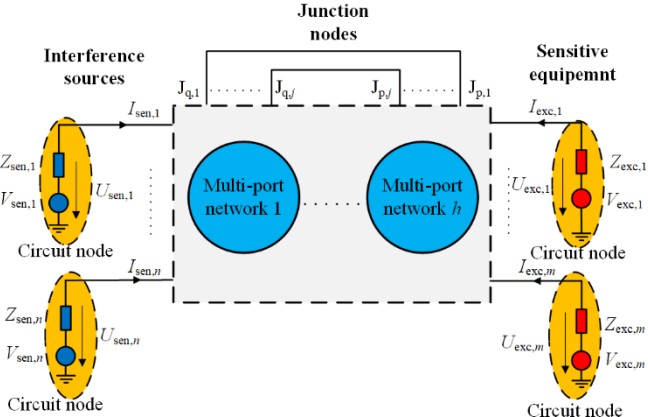

**Figure 3.** Global multi-port network.

In Figure 3, the dotted box represents the global multi-port network containing $h$ sub-networks indexed by $1, \cdots , h$ one by one. It should be noted that here "Interference sources" refers to the parts whose equivalent voltage is not zero and "Sensitive equipment" refers to the parts with zero equivalent voltage. The direct connection ports among different

sub-networks are denoted by $J_{q,i}$ and $J_{p,i}$ ($i = 1, \cdots, j$). For each multi-port network, it satisfies [10]:

$$U_i = Z_i \cdot I_i \qquad (1)$$

where $Z_i$ is the input impedance of the network, and $U_i = \begin{bmatrix} U_{i,1} \\ U_{i,2} \\ \vdots \end{bmatrix}$ and $I_i = \begin{bmatrix} I_{i,1} \\ I_{i,2} \\ \vdots \end{bmatrix}$ are the port voltage and port current, respectively. Here the Z-parameter of multi-port network is selected because it is independent of the port impedance.

For each circular node, we have:

$$V_i - U_i = Z_{\text{in},i} \cdot I_i \qquad (2)$$

where $V_i$ is the equivalent voltage of the circuit node and $Z_{\text{in},i}$ is a diagonal matrix whose diagonal elements are the port impedances. We have the following equations for each multi-port network by combing Equations (1) and (2) together:

$$I_i = (Z_i + Z_{\text{in},i})^{-1} \cdot V_i, \quad U_i = Z_i \cdot (Z_i + Z_{\text{in},i})^{-1} \cdot V_i. \qquad (3)$$

To construct the equation of the global network, the variables in Equation (3) is re-arranged according to their types, i.e., "Interference sources", "Sensitive equipment" or direct connections. The following vectors are defined to describe the global network:

$$U = \begin{bmatrix} U_{\text{sen}} \\ U_{\text{exc}} \\ U_q \\ U_p \end{bmatrix}, \quad I = \begin{bmatrix} I_{\text{sen}} \\ I_{\text{exc}} \\ I_q \\ I_p \end{bmatrix}, \quad V = \begin{bmatrix} V_{\text{sen}} \\ V_{\text{exc}} \\ V_q \\ V_p \end{bmatrix}, \qquad (4)$$

where the subscript denotes the types of ports with "sen" for "Sensitive equipments", "exc" for "Interference sources", "q" and "p" for the junction nodes $J_q$ and $J_p$, respectively. The vectors, $U_{\text{sen}}$, $I_{\text{sen}}$ and $V_{\text{sen}}$, are composed of $U_{\text{sen},i}$, $I_{\text{sen},i}$ and $V_{\text{sen},i}$, $i = 1, \cdots, n$ with the same order. Other vectors in Equation (4) are defined similarly. Then being similar to Equation (4), the parameter of the global network is also re-arranged and defined as:

$$Z = \begin{bmatrix} Z_{\text{sen,sen}} & Z_{\text{sen,exc}} & Z_{\text{sen},q} & Z_{\text{sen},p} \\ Z_{\text{exc,sen}} & Z_{\text{exc,exc}} & Z_{\text{exc},q} & Z_{\text{exc},p} \\ Z_{q,\text{sen}} & Z_{q,\text{exc}} & Z_{q,q} & Z_{q,p} \\ Z_{p,\text{sen}} & Z_{p,\text{exc}} & Z_{p,q} & Z_{p,p} \end{bmatrix}, \quad Z_{\text{in}} = \text{diag}\begin{pmatrix} Z_{\text{sen,in}}, & Z_{\text{exc,in}}, & 0_j, & 0_j \end{pmatrix} \qquad (5)$$

where $0_j$ denotes the $j$-dimensional zero matrix, $\text{diag}(.)$ denotes the diagonal matrix taking the parameter as its diagonal elements, $Z$ and $Z_{\text{in}}$ are the input impedance and port impedance of the global network, respectively. With the parameters and variables defined by Equations (4) and (5), the following function can be obtained according to the multi-port network theory of Equation (10):

$$U = Z \cdot \begin{bmatrix} G_{\text{sen}} \cdot Z + G_{\text{sen}} \cdot Z_{\text{in}} \\ G_{\text{exc}} \cdot Z + G_{\text{exc}} \cdot Z_{\text{in}} \\ G_U \cdot Z \\ G_I \end{bmatrix}^{-1} \cdot V \qquad (6)$$

where $G_U = \begin{bmatrix} 0 & 0 & E_q & -E_p \end{bmatrix}$ and $G_I = \begin{bmatrix} 0 & 0 & E_q & E_p \end{bmatrix}$ are used to make the "Junction nodes" in Figure 3 follow the Kirchhoff's law, and $G_{\text{sen}} = \begin{bmatrix} E_{\text{sen}} & 0 & 0 & 0 \end{bmatrix}$ and $G_{\text{exc}} = \begin{bmatrix} 0 & E_{\text{exc}} & 0 & 0 \end{bmatrix}$ are used to select the parameters corresponding to "Sensitive equipments" and "Interference sources" from the global network parameter, respectively. In the aforementioned matrices, $E_\#$ denotes the unit matrix whose dimension equals the number of ports belonging to the type denoted by its subscript, which is the same as the definition in Equation (4).

Equation (6) gives out the algebraic function to calculate the port voltage. Similarly, other variables, such as port current, induced voltage and current in the equivalent circuit nodes, and etc. can also be calculated.

## 4. Model-Based Analysis and Improvement

Even with a precise model, it is still difficult to determine the final engineering solution because:

(a)   A vehicle-level radiation problem may be caused by multiple factors simultaneously;
(b)   The determination of the final solution also depends on such untechnical factors as application difficulties and costs, which are hard to evaluate quantitatively.

In this section, a systematic approach based on the topological model set up in Section 3 is presented to realize improvement by analysis of the importance of technical factors with EWM [17,18] and evaluation of both technical and untechnical factors together using AHP [19,20].

### 4.1. Importance Analysis

To introduce the importance analysis of technical factors succinctly, it is assumed that the interferences related to all "Interference sources" and "Sensitive equipment" are described in the form of voltage. For each factor, its influence on the EMC problem is evaluated from two aspects, i.e., contribution degree and sensitivity. The former reflects the contribution on the interference voltage and is calculated by:

$$\Delta_i = U - U_{r,i} \tag{7}$$

where the subscript $i$ is the index of the considered technical factor, $\Delta_i$ is the contribution degree, $U$ and $U_{r,i}$ are the original interference voltage and that calculated by Equation (6) when the factor $i$ is set to be zero, respectively. A larger $\Delta_i$ implies that this factor contributes more on the radiated emission and it should be improved technically.

The contribution degree is established with the assumption that each factor can be attenuated totally, which is impractical in engineering. So besides the contribution, another index called sensitivity is introduced to evaluate the improvement potential of the technical factor:

$$S_i = \partial U / \partial F_i \tag{8}$$

where the subscript $i$ is the index of technical factor whose sensitivity is $S_i$, and $F_i$ denotes the considered factor including the interference voltage of sources and coupling coefficients in the network. Essentially, sensitivity indicates the effect by improving the factor with unit quantity at the current state.

To comprehensively consider these two totally different indexes, EWM is adopted to measure the importance of each factor:

$$S_{c,i} = \omega_1 x_{i1} + \omega_2 x_{i2}$$
$$x_{i1} = \frac{\Delta_i - \min_i \Delta_i}{\max_i \Delta_i - \min_i \Delta_i}, \quad x_{i2} = \frac{S_i - \min_i S_i}{\max_i S_i - \min_i S_i} \tag{9}$$

where $S_{c,i}$ is the evaluation score of factor $i$ which measures the importance, $x_{i1}$ and $x_{i2}$ are the Min-Max normalized values of the contribution degree and sensitivity, respectively. The weight coefficient, $\omega_j$, is calculated as the following by EWM [17]:

$$\omega_j = \frac{1 + \frac{\sum_{i=1}^{n}\left(P_{ij} \cdot \log_{10} P_{ij}\right)}{\ln n}}{\sum_{j=1}^{2}\left[1 + \sum_{i=1}^{n}\frac{\left(P_{ij} \cdot \log_{10} P_{ij}\right)}{\ln n}\right]}, \quad P_{ij} = x_{ij} / \sum_{i=1}^{n} x_{ij} \tag{10}$$

A larger evaluation score implies that technically the corresponding factor contributes more on the radiated emission.

*4.2. Improvement Solution Selection*

The importance analysis only can find out the key technical factor, but other untechnical factors are also needed to be considered for the selection of the final solution, such as application difficulties, costs and etc. Compared with the technical factors, this is even more difficult because most of these untechnical factors only can be evaluated subjectively. To determine the final solution more effectively and efficiently, a hierarchical evaluation system as shown in Figure 4 is proposed.

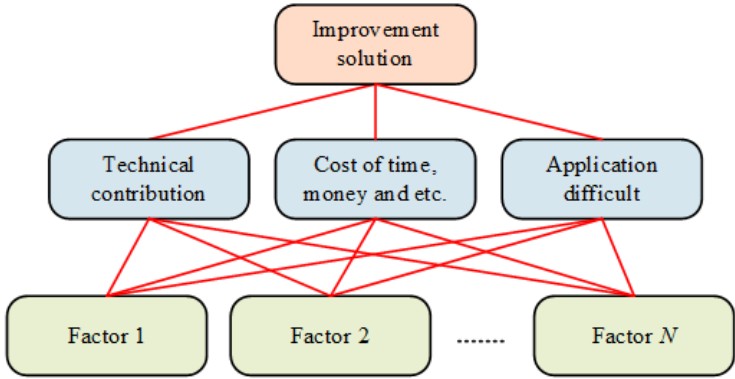

**Figure 4.** Hierarchical evaluation of improvement solution.

In this hierarchical structure, the improvement solution is evaluated from three aspects, i.e., technical contribution, cost and application difficulty. The technical contribution is measured quantitatively using the importance analysis results described by Equation (9). Other aspects are subjective and difficult to be evaluated quantitatively. To make full use of the engineer's experience, meanwhile eliminate the influence of subjectivity as far as possible, the Delphi method is adopted [19–21]. By this approach, the elements at the same layer are evaluated comparatively and can be graded by the nine-points method (1 for lowest priority and 9 for highest priority).

The judgement matrix of the first layer is defined as $J_1 \in \mathbb{R}^{3 \times 3}$, whose normalized eigenvector corresponding to the maximum positive eigenvalue is $\phi_1^* \in \mathbb{R}^3$ [22]. Similarly, the judgement matrices of technical contribution, cost and application difficulty are $J_{2,i}, i = 1, \cdots, 3$, $J_{2,i} \in \mathbb{R}^{N \times N}$. Their normalized eigenvectors are $\phi_{2,i}^* \in \mathbb{R}^N$. Accordingly, the final evaluation score is:

$$\phi_c^* = \begin{bmatrix} \phi_{21}^* & \phi_{22}^* & \phi_{23}^* \end{bmatrix} \phi_1^* \tag{11}$$

where $\phi_c^*$ is the final evaluation score. A bigger value in $\phi_c^*$ implies that the corresponding factor is preferred to be selected and the corresponding improve measurement tends to be the final solution.

## 5. Application and Analysis

In Sections 3 and 4, a systematic approach is proposed to model, diagnose and improve the vehicle radiation problem. To realize this process in practice, some specific methods or techniques are required. In this section, the proposed strategy is adopted to diagnose and improve of a radiation emission problem of an EV in the frequency range from 15 kHz to 30 MHz to show some details for example and validate its effectiveness. The radiation emission is measured according to SAE J551-5 [23] and some results are shown by Figure 5. The experiment is conducted in the 10 m semi-anechoic chamber, the electric field and magnetic field are measured by the rod and loop antennas respectively, which are located 1 m away from the vehicle body. During the testing, the vehicle running state and test system layout are consistent with the regulation, the electric field is measured with vertical polarization, but the magnetic field is measured with three antenna orientations. It is found that, at around 13.85 MHz, the electric field exceeds the limit by about 7 dBμV/m.

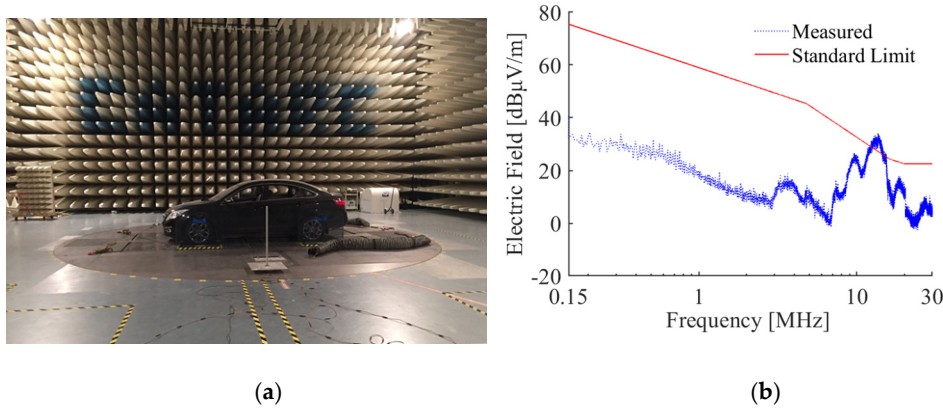

(**a**)　　　　　　　　　　　　　　　　　　　　　　　(**b**)

**Figure 5.** Radiated emission test: (**a**) Test site; (**b**) Electric field.

Since the vehicle layout is delivered and fixed, it is almost impossible to change the component arrangement and the design of the electrical system. A feasible way is to find out and improve the main interference source. One traditional way to identify the source is by comparing the emission under different combinational running states by switching on/off different components. This is very resource-consuming and cannot provide an efficient improvement measure. Here, the systematic approach presented in Sections 3 and 4 is adopted to identify the main interference source and determine the final solution.

### 5.1. Modeling of Radiation Emission

Normally, the radiated emission at low frequency range is caused by the high voltage systems including PDU, compressor, battery, DC/DC, on board charger (OBC), positive temperature coefficient heater (PTC), battery heater, driver inverter and motor. The schematic of these systems is shown in Figure 6a,b. It should be noted that the indexes in Figure 6 are defined for the convenience of modelling for this radiation problem and are different from those in Figure 2.

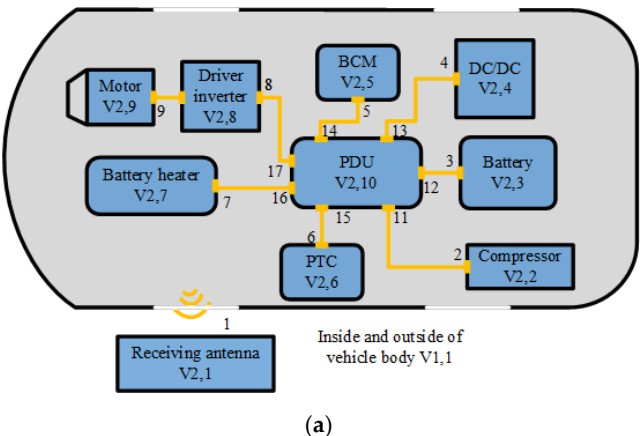

(**a**)

**Figure 6.** *Cont*.

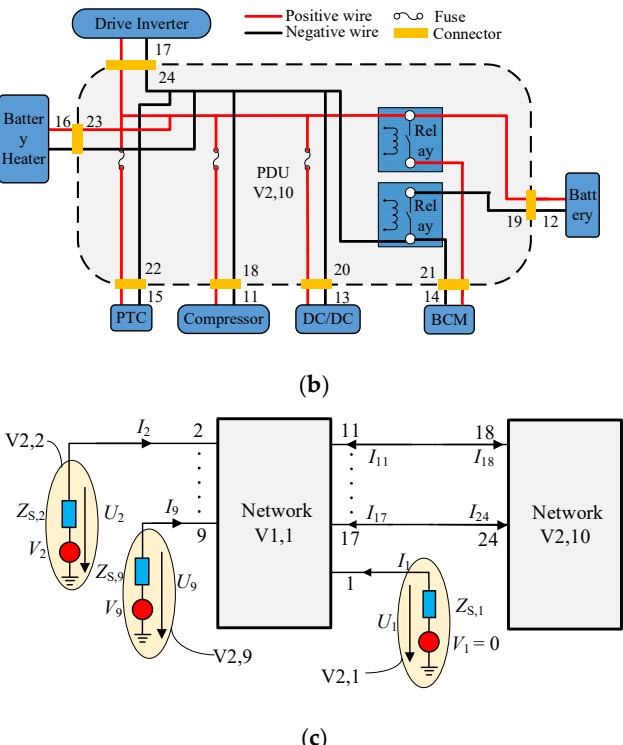

**(b)**

**(c)**

**Figure 6.** Modelling process: (**a**) Schematic diagram of power system; (**b**) Schematic diagram of PDU; (**c**) Topological multi-port network diagram.

According to the topological modelling approach introduced in Section 3 and the test requirement of SAE J551-5, a topological model composed of two networks is established as show by Figure 6c considering the following facts:

(1)　The positive and negative wires are arranged in parallel and the loop area of the differential mode interferences is comparatively smaller than that of common ones. In the present modelling, only the common mode interferences are considered and the differential ones are neglected;

(2)　All mentioned power components in Figure 6a are packed in the metal shells, which are finely grounded;

(3)　PDU is a junction of all power cables and its internal schematic is shown by Figure 6b.

In Figure 6c, the network V1,1 is to describe the propagation of electromagnetic field in/out the vehicle space and the other is for the coupling in PDU. When modelling the nodes, their equivalent impedances and voltages are assumed to be independent of the working state. To ensure the accuracy, the voltage $U_i$ and the current $I_i$ are by the oscilloscope and current probe under the same running condition as SAE J551-5. With the measured voltage and current, the equivalent voltages of nodes connected to the network are calculated by:

$$V_i = U_i + Z_{S,i} I_i \tag{12}$$

where $U_i$ and $I_i$ are the measured voltage and current, $Z_{S,i}$ is the impedance measured by the network analyzer when power off. Some of the results are shown by Figure 7.

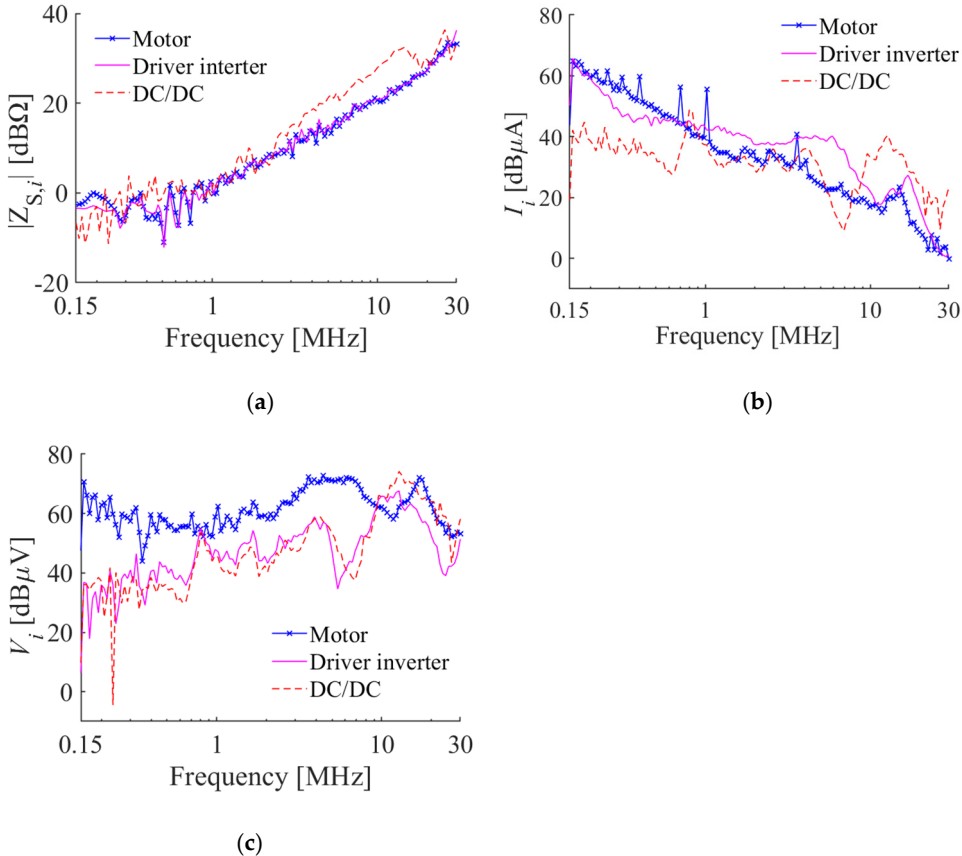

(a)

(b)

(c)

**Figure 7.** Parameters of equivalent circuit nodes: (**a**) Equivalent internal impedance; (**b**) Interference current; (**c**) Equivalent interference voltage.

The parameter of networks is calculated from the S-parameter, which is solved numerically by FEKO [24] as shown in Figure 8. For the feasibility and efficiency of computation, the model was simplified by deleting the nonmetal components of vehicle body, comparatively small metal structures and electric components, and the low voltage wiring harness. These two models are solved separately to reduce the consumption of time and computation resources. Moreover, several methods, such as experimental test, numerical calculation, etc., can also be used to acquire the data required by Equation (6).

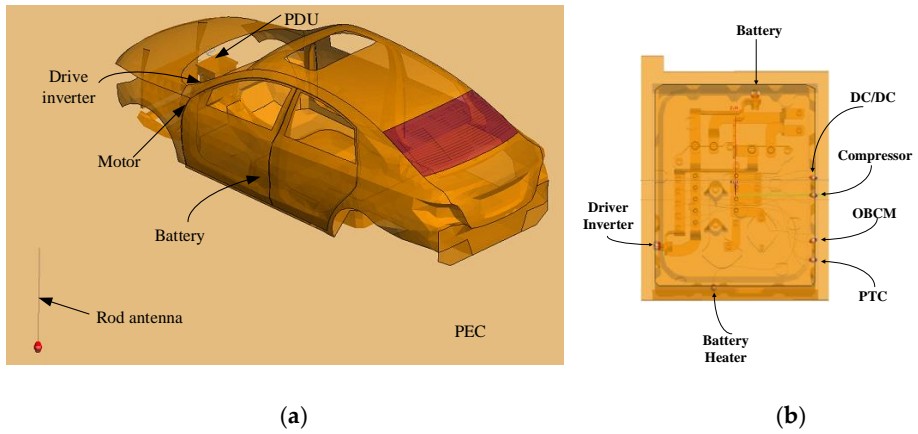

(a)

(b)

**Figure 8.** Simulation model in FEKO: (**a**) EV model; (**b**) PDU model.

Some of the simulation results of the S-parameter and the calculated Z-parameter are shown in Figure 9, where the subscript of variables denotes the index of ports.

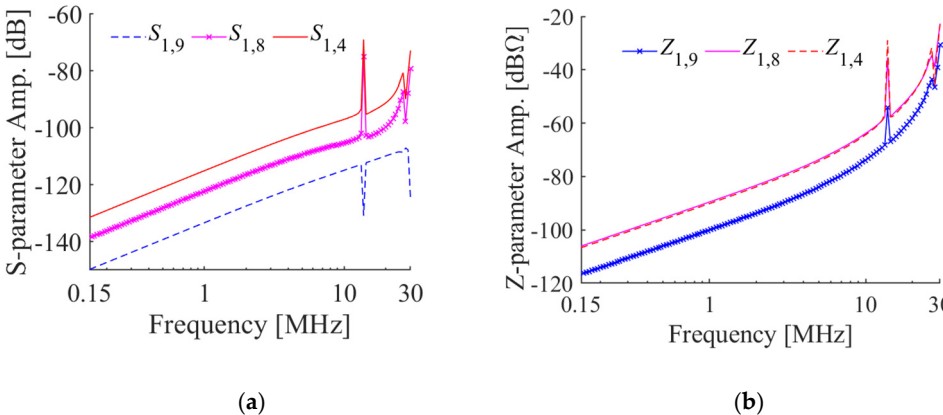

**Figure 9.** Parameters of networks: (**a**) S-parameter; (**b**) Z-parameter.

The radiation emission predicted by the topological model is compared with the experimental one in Figure 10. Overall, they have the same trend and the frequency range where the electric field exceeds the limit is similar. Though the simulation result has some small peaks in the frequency range between 1 MHz and 3 MHz, the electric field in this frequency range is much lower than the limit. When using this model to diagnose the emission problem, we focus on the frequency range where the emission is over the limit. The comparative results show that this model can be used to find the main factor causing the emission problem.

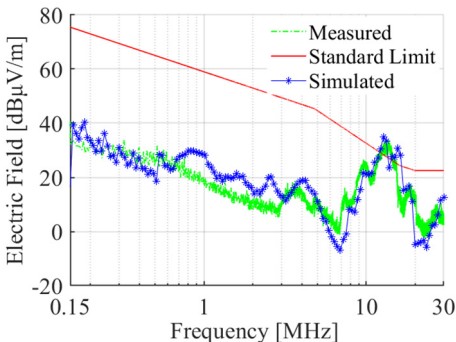

**Figure 10.** Comparing result of electric field at left side.

### 5.2. Problem Analysis and Improvement

As shown in Figure 5, the electric field is out of the limit. To identify the main technical factor causing this emission problem. The importance analysis of each factor has been conducted using the method introduced in Section 4.1 and some analysis results of the contribution and improvement potential of technical factors are shown in Figure 11.

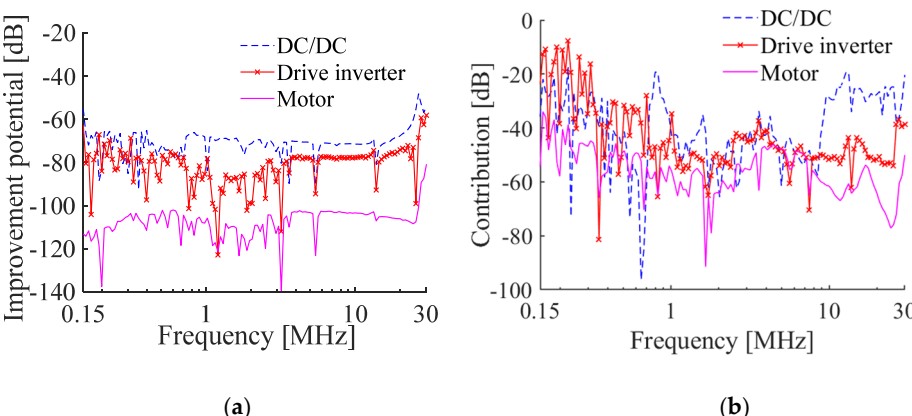

**Figure 11.** Technical analysis results: (**a**) Improvement potential; (**b**) Contribution.

The comprehensive evaluation score calculated by Equation (9) are visualized by the heatmap shown in Figure 12a, where $S_i$ ($i = 1\sim3$) denotes the interference sources, $P_i$ ($i = 1, 2, 3, \dots$) denotes the coupling paths. According to the evaluation score, the DC/DC converter (denoted by S3) is the main interference source technically. But other non-technical factors, such as application difficulty and cost, are also should be considered to determine the final solution. It is almost impossible to measure these non-technical factors quantitatively and accurately. With the technical evaluation score shown in Figure 12a, five experts are invited to conduct the hierarchical analysis to get the judgement matrices, $J_1$ and $J_{2,i}, i = 1, \cdots, 3$. As shown by Figure 4, $J_1$ is used to evaluate the relative importance among the three factors in the first layer, i.e., cost, application difficulty and technical contribution. The judgement matrices $J_{2,i}$ are for the evaluation of the factors in the third layer. For each factor, its importance is evaluated by compared with other factors in the same layer and measured from 0 to 9 (0 is the least important one and 9 is most important). With the judgement matrices, the evaluation score to determine the final solution can be calculated by Equation (11) and the results are shown in Figure 12b.

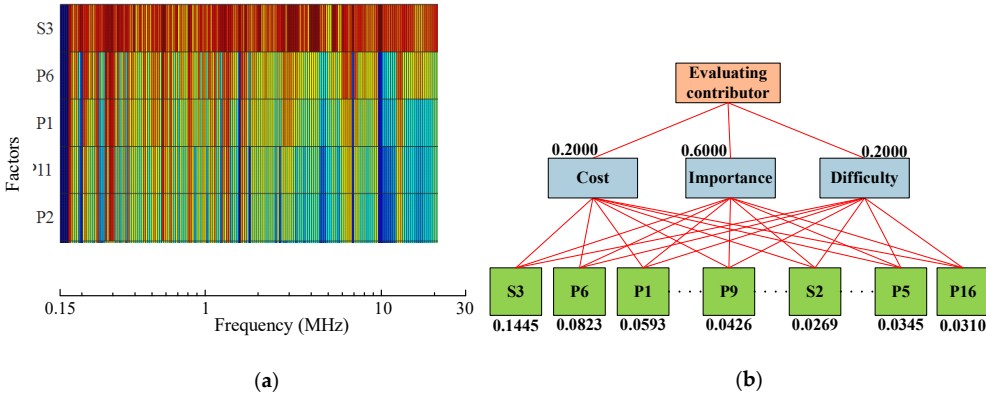

**Figure 12.** Problem analysis results: (**a**) Importance analysis results; (**b**) Hierarchical analysis for final solution.

According to the hierarchical analysis result, DC/DC (denoted by S3) and its coupling path with the measurement antenna (denoted by P6) lie in the top two. Their final evulation scores are 0.1445 and 0.0823, respectively. The interference generated by DC/DC is preferred to be attenuated, and considering the frequency range where the electric field exceeds the limit, a single-phase DC power filter is selected finally.

The radiated electric field with the selected filter is shown by Figure 13 which satisfies the requirement of SAE J551-5. Though the proposed strategy has been applied to diagnosis and improvement of the radiation problem of EV at the stage when the sample vehicle

is ready, it can also be adopted at the early stage if the required model parameters are available.

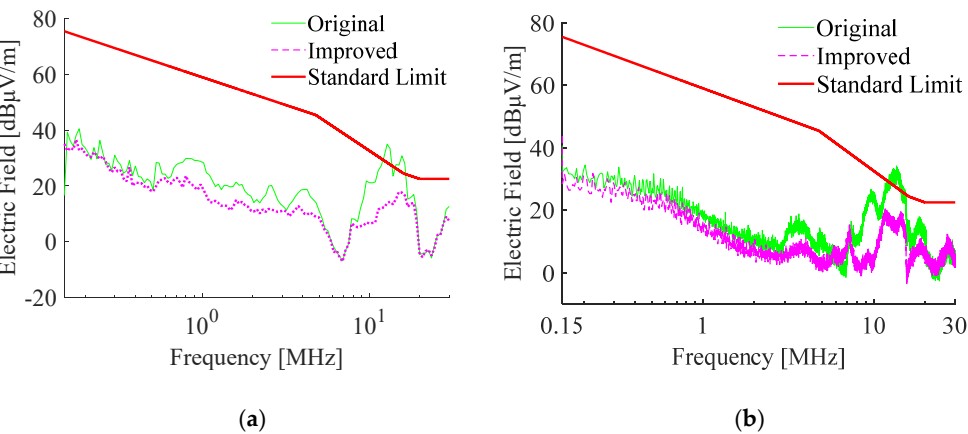

**Figure 13.** Improvement results of electric field at left side: (**a**) Simulation; (**b**) Experimental test.

Summarizing the model based analysis and improvement process for vehicle radiation problems introduced in Sections 3 and 4, and the application results in this section, the advantages and disadvantages of the presented strategy are summarized in Table 1.

**Table 1.** Advantages and disadvantages of model based analysis and improvement strategy.

| Advantages | Disadvantages |
| --- | --- |
| <ul><li>Modeling difficulty is reduced</li><li>Network parameters can be obtained by a variety of methods independently</li><li>Less engineering experience is required</li><li>Both technical and untechnical factors are considered comprehensively</li></ul> | <ul><li>Circuit nodes should be linear and their characteristics should be independent of working states</li><li>Coupling paths in networks should be linear and passive</li><li>Some engineering experiences still are required to do the hierarchical analysis</li></ul> |

## 6. Conclusions

This paper proposes a systematic method to solve vehicle-level radiation problems including the topological modelling approach, model-based analysis of technical factors and determination of engineering solutions considering untechnical factors such as cost and application difficulty. The application results show that the proposed strategy has the following advantages:

(a) By introducing the topological modeling method, the separate modeling and simulation can be achieved. Hence the faster simulation speed, the lower hardware requirement, and more accurate as well as more development efficient.

(b) The influences of technical factors on the radiation problem can be evaluated quantitatively from the aspects of sensitivity and contribution by EWM.

(c) The proposed hierarchical evaluation method can combine untechnical factors, i.e., application difficulty and cost, with the technical ones together to determine the final solution for the radiation problem.

Some open questions about the detail problems are worthy of further investigation as follows:

(a) In this study, it is assumed the equivalent impedance of the circuit node is independent of its working state. This is no long valid if the circuit has high nonlinearities. If nonlinear components are used to describe the node, the equivalent interference cannot be solved by Equation (6) directly. Some analysis method for nonlinear circuits is required and need to be further studied;

(b) Moreover, the equivalent impedance of the circuit node is measured under the state of power off. When the circuit is activated, how to measure the equivalent impedance is still a problem especially for the high power switching circuit;

(c) During the improvement process, it is possible to consider the effectiveness of the potential solutions quantitatively with the topological model;

(d) In this study, only the common mode interference is considered and how to model the differential mode ones need to be further studied.

**Author Contributions:** F.G. wrote the paper and provided the main idea, Q.W. conducted the data analysis and modelling of EMI, and Y.X. provided the experimental data. All authors have read and agreed to the published version of the manuscript.

**Funding:** This work was supported in part by the Open Fund of Chongqing EMC Technology Research Center under the grand 19Akc6 and National Key R&D Program of China under grant 2017YFB0102504.

**Institutional Review Board Statement:** Not applicable.

**Informed Consent Statement:** Not applicable.

**Data Availability Statement:** The study did not report any data.

**Conflicts of Interest:** The authors declare no conflict of interest.

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
