# Peer review of "Model-Based Analysis and Improvement of Vehicle Radiation Emissions at Low Frequency"

_applsci, doi:10.3390/app11178250_

Round 1
Reviewer 1 Report
The paper is of sufficient interest for the reader. Some minor issues should be addressed before publication:
1) Figure 5, even if already present in the standard, please provide a brief description of the test set-up: chamber dimension, type of antenna, distance from the car, etc.
2) At row 342, "Here, the systematic approach presented in section 0 and 0 ", please clarify "0 and 0"
3) The same at row 359.
4) Rows 362 and 363. "The positive and negative wires are arranged in parallel, and so only the common mode interferences are considered" This is not completely true. Even if in parallel, the produce a differential mode radiation, which is in general lower at lower frequencies. Please comment this aspect and add a sentence to state that "differential mode radiation is neglected in the present modelling."
5) In the conclusions, where you describe the open questions, ad an item to specify that the differential mode radiation modelling is another open question
Author Response
Please find the response in the attachment.

Reviewer 2 Report
The paper combines the idea of topological decomposition for electromagnetic problems with a means of evaluating/deciding the best solution. Results are presented to demonstrate the utility of the method.
Overall the paper is clear and well written apart from some minor grammatical issues. I have marked detailed comments on the manuscript.
The paper would benefit from the citation of previous authors that have considered topological solutions and some more explanation of how the empirical factors were derived and used in the results section.

Author Response
Please find the answer in the attachment.

Reviewer 3 Report
This article is assumed the equivalent impedance of the circuit node is independent of its working state. The applications already limitation. The overall parameters of topological modeling method have not consider at 10~20 MHz. If possible, please list solutions to comparison
Author Response

(The authors gave the same response as above.)

Round 2
Reviewer 3 Report
The proposed results have not any root cause analysis, that is only problems analysis and hierarchical analysis. The model diagnosis condition must list advantage and disadvantage, how about comparison table?
Round 3
Reviewer 3 Report
The proposed article must summary performance comparison with detail solutions.
Author Response
It is very sorry that we cannot understand the exact meaning of your comment.
In the manuscript, we have added the advantages and disadvantages of the proposed strategy (See Tab. 1). For concise, some typical results of the intermediate processes are also shown (See Fig.7, Fig. 9 and Fig. 11). The prediction result by the topological model has been compared with the experiment one to validate the accuracy of the model.
Please give us some detailed and specific suggestions.